# Online Sign Identification: Minimization of the Number of Errors in Thresholding Bandits

**Reda Ouhamma**
Univ. Lille, Inria, CNRS,
Centrale Lille, UMR 9189 CRIStAL,
F-59000 Lille, France
`reda.ouhamma@gmail.com`

**Rémy Degenne**
Univ. Lille, Inria, CNRS,
Centrale Lille, UMR 9189 CRIStAL,
F-59000 Lille, France
`remy.degenne@inria.fr`

**Pierre Gaillard**
Univ. Grenoble Alpes, Inria, CNRS,
Grenoble INP, LJK, 38000 Grenoble, France
`pierre.gaillard@inria.fr`

**Vianney Perchet**
Crest, Ensae & Criteo AI Lab
`vianney.perchet@normalesup.org`

## Abstract

In the fixed budget thresholding bandit problem, an algorithm sequentially allocates a budgeted number of samples to different distributions. It then predicts whether the mean of each distribution is larger or lower than a given threshold. We introduce a large family of algorithms (containing most existing relevant ones), inspired by the Frank-Wolfe algorithm, and provide a thorough yet generic analysis of their performance. This allowed us to construct new explicit algorithms, for a broad class of problems, whose losses are within a small constant factor of the non-adaptive oracle ones. Quite interestingly, we observed that adaptive methods empirically greatly out-perform non-adaptive oracles, an uncommon behavior in standard online learning settings, such as regret minimization. We explain this surprising phenomenon on an insightful toy problem.

## 1 Introduction and related work

In a stochastic multi-armed bandit problem, a decision maker sequentially samples from different distributions in order to optimize a loss that depends on the unknown parameters of those distributions. As a consequence, a tradeoff arises between gathering more samples from any possible distribution (to enhance the estimation of relevant parameters) and optimizing the allocation to minimize the final loss. We can distinguish two main categories of losses, focusing on "exploitation" vs "exploration". The former directly depends on the whole allocation of samples and the typical example is regret minimization (we refer to the recent monographs [27, 6, 34] that cover this setting almost exhaustively). The later is a bit different; after the budget of samples is exhausted, the algorithms must answer one or several "questions" (on the different distribution) and its loss is related to the number of mistakes made; the typical application being best-arm identification and variants [3, 24].

We investigate a class of pure exploration problems, called "thresholding bandit" [28, 35]. The key property of this class is that a question is asked about each distribution, and the probability of making a mistake decreases with the total information gathered on that distribution solely. The typical question the algorithm must answer is "is the mean of the distribution above or below some threshold?" (say, 0, for simplicity); giving the wrong answer can either incur a unit cost - independently from the distribution -, or a data-dependent cost (say, the distance to the threshold that represents the "risk" of that distribution). A typical application of thresholding bandits is crowdsourcing [7] where the

35th Conference on Neural Information Processing Systems (NeurIPS 2021).

objective is to distinguish workers with positive (vs. negative) efficiency; another one is bandit binary classification [18].

Some care must be taken when designing a performance criterion for a thresholding bandit problem, since any non-stupid algorithm will eventually answer all questions correctly (hence have a 0 loss) if it has enough samples. Furthermore, if distributions are sub-Gaussian (a rather mild assumption that we are going to make), the probability of making a single mistake decreases exponentially fast with the number of samples. As a consequence, the focus must be on controlling the exponential decay constant. We illustrate that issue on the unit cost problem described as follows. There are $K$ different $\sigma$-sub-Gaussian distributions; the mean of distribution $k$ is denoted by $\mu_k$ and the (variance-normalized) gap of distribution $k$ to the threshold $0$ is denoted by $\Delta_k := |\mu_k|/\sqrt{2\sigma^2}$. The algorithm has a budget of $T$ samples to (sequentially) allocate to those distributions and, based on the $N_{k,T}$ samples of distribution $k$, it must decide the sign of $\mu_k$; any mistake has a cost of one. We denote by $E_k \in \{0,1\}$ an indicator of a wrong sign prediction of $\mu_k$ after exhausting the budget of $T$ samples. The loss is then $L_T^1 := \sum_k E_k$. It is not difficult to see that the expected number of mistakes could be of order $\sum_{k=1}^K \exp(-N_{k,T}\Delta_k^2)$ .

In particular, sampling evenly across distributions ($N_{k,T} = T/K$) gives $\mathbb{E}[L_T^1] \approx \sum_k \exp(-\frac{T}{K}\Delta_k^2)$, which has an exponential decay in $T$. However, this uniform allocation is far from being optimal in term of the exponential decay constant. Computing an (approximate) optimal fixed allocation in hindsight is not difficult: just optimize the upper-bound of $\mathbb{E}[L_T^1]$. Since even the uniform allocation has a loss decaying exponentially, the performance of an algorithm should be measured not with respect to $\mathbb{E}[L_T^1]$ (see [24]) but rather in terms of $-\log(\mathbb{E}[L_T^1])/T$. The oracle that uses knowledge of the gaps $\Delta_k$ to optimize its fixed allocation verifies

$$\limsup_{T\to\infty} \frac{1}{T}\log(\mathbb{E}[L_T^1]) \leq -\frac{1}{\sum_k 1/\Delta_k^2} \ .$$

This unit cost framework has been investigated recently [35] with a simple yet effective algorithm called LSA (Logarithmic-Sample Algorithm) designed exclusively for this problem; it samples the distribution with the smallest current index defined as $\alpha N_{k,t}\hat{\Delta}_{k,t}^2 + \log N_{k,t}$, where $\hat{\Delta}_{k,t}$ is the empirical estimate of $\Delta_k$ and $\alpha$ is some parameter to be chosen. LSA is "optimal up to a constant", but the constant is unfortunately in the exponential decay, as it was proved that[1]

$$\limsup_{T\to\infty} \frac{1}{T}\log(\mathbb{E}[L_T^1]) \leq -\frac{1}{16020}\frac{1}{\sum_k 1/\Delta_k^2} \quad \text{for LSA.}$$

As we shall see, this result can be drastically improved with our more refined and general analysis (that implies choosing a totally different input parameter $\alpha = 1$ instead of $1/10$ as suggested originally).

## 1.1 Contributions

We investigate the thresholding bandit problem with a weighted number of errors loss. Our contributions are twofold: 1) a generic method to design algorithms, with a generic proof, showing good performance on the weighted number of errors loss. 2) new lower-bounds and counter-intuitive results for the unit cost problem.

**A generic algorithm with performance guarantees** We propose a Frank-Wolfe inspired method to design bandit algorithms. We develop a proof technique to obtain loss bounds for the type of algorithms that our method produces, which we apply to the thresholding bandit with losses

$$L_T = \sum_{k=1}^K a_k E_k \quad \text{or} \quad L_T^\Delta = \sum_{k=1}^K \Delta_k E_k \ , \tag{1}$$

where $(a_k)_{k\in[K]}$ are known costs. The class of algorithms we analyze includes both LSA and APT (Anytime Parameter-free Thresholding) [35, 28]. We obtain precise non-asymptotic loss bounds for $\mathbb{E}[L_T]$; for instance, we improve the original bound of LSA by a factor 4005 (and APT by a factor 8). More importantly, we get a new algorithm whose expected error for the unit cost problem is within a

---

[1]See Remark 1 [35]. This bound implies that LSA - with the specified choice of $\alpha = 0.1$ needs 16000 times more samples than the oracle to achieve the same performances.

factor 4 of the oracle. We emphasize again than those "constant" factors are in the exponential (and are not mere multiplicative constants).

Interestingly, this class of algorithms are *not* driven either by the "optimism under uncertainty" principle, a standard technique in multi-armed bandit [4] nor "Explore-then-commit / Successive Elimination" [31, 12].

**New insights on the thresholding bandit problem** First, the optimal allocation provided by the oracle of [35] in the unit cost problem has a M-shape (see Figure 1) because of two concurrent phenomena. On the one hand, the arms close to the threshold should not be pulled too much because their sign is difficult (if not impossible) to identify and it is a waste of budget. On the other hand, the signs of the arms far from the threshold are quickly well estimated and therefore should not be chosen too often either. The middle arms are the ones that need to be pulled the most frequently. As $T$ gets larger, more and more budget is allocated to difficult arms. In section 2.2, we provide a lower-bound that shows that this M shape is actually impossible to achieve for a sequential algorithm. Typically, the hollow inside of the M shape corresponds to arms whose sign cannot be well-estimated. In particular, it is not possible to distinguish arms that are very close to the threshold from the arms that are at the top of the M and should be pulled the most frequently according to the oracle.

Our second insight is corroborated by numerical simulations in Section 4. We show empirically that our algorithms not only match but also surpass the optimal non-adaptive sampling of the oracle. We conjecture that our algorithms take advantage of the chance due to noise that can move its estimate of the arm away from the threshold. In particular, when all the gaps $\Delta_k$ are equal, the non-adaptive optimal allocation should be uniform, which is significantly outperformed by adaptive algorithms. This suggests that adaptivity is crucial for this problem and may inspire future research directions to the multi-armed bandit community in order to prove theoretical guarantees for such phenomena.

## 1.2 Additional related work

**Zero-one loss** Most of the literature on thresholding bandits [28, 30, 8] aims at minimizing the probability of making any sign error, i.e., minimizing the loss

$$L_T^* = \mathbb{I}\{\exists k \in [K],\ E_k = 1\} = \max_k E_k. \tag{2}$$

We already mentioned the algorithm APT [28], that gets an exponential decay of that loss (variants include variance estimation [36] and/or delayed feedbacks). Other algorithms exist, but based on the optimism principle [23, 30]. Unfortunately they suffer from a degraded exponential decay constant (by a factor bigger than 1000).

Another part of the literature focuses on the fixed confidence framework, where the objective is to answer some questions with some fixed probability of mistake (and obviously with a minimal sample budget). For instance, an objective could be to return any arm above some threshold as soon as possible [21, 9], or the one closest to the threshold [16], or just identifying that one arm is above that threshold [25], or even to control false discovery rates and variants [19, 18].

**Global loss, dynamic allocation and outliers detection** The loss considered in thresholding bandits can be seen as a variant of a "global loss" (i.e., essentially non-linear) that has been extensively studied in the bandit literature [1, 2, 29]. However, the major difference is, again, that the optimal allocation is time dependent and that the loss converges exponentially fast to zero (no matter the algorithm). Similarly, Frank Wolfe algorithms have been introduced in this setting [5, 13]; even though our algorithms share some similarities, they are intrinsically different for the same reasons.

Similarly, the problem investigated could be seen as a special case of bandit resource allocations [26, 7, 32, 11, 14] but where the loss is always decreasing with respect to the budget allocated per resource (hence again leading to a zero loss exponentially fast).

Finally the global objective of thresholding bandits is to obtain a synthetic view of how the means of distributions are spread on the real line (which ones are above/below some threshold). In that aspect, this problem sheds some similarities with outlier detection in multi-armed bandits [22, 38, 37].

## 2 Preliminaries

We describe here the weighted number of errors setting, in which an error on arm $k$ has a known cost $a_k > 0$. The sum-of-gaps setting will be briefly investigated in section 3.3. The environment is composed of $K > 1$ arms and an algorithm sequentially pulls them. After pulling arm $k \in [K]$, it observes a sample from a distribution $\nu_k$ with mean $\mu_k$, and that sample is independent of past observations. The distribution $\nu_k$ is supposed $\sigma$-sub-Gaussian, that is

$$\forall \lambda \in \mathbb{R} : \mathbb{E}_{X \sim \nu_k} \left[ \exp(\lambda(X - \mu_k)) \right] \leq \exp(\sigma^2 \lambda^2 / 2) .$$

The total number of rounds (and samples) $T$ is known in advance and called the horizon. After pulling $T$ arms, the task of the algorithm is to classify the arms depending on whether $\mu_k > \theta$ or not, where $\theta$ is a known threshold that we conveniently set to 0 (although it could be any other value, even different from arm to arm, without significant change to the analysis). Let $s_k \in \{-1, 1\}$ be the sign of $\mu_k - \theta$, equal to 1 iff $\mu_k - \theta > 0$. The algorithm returns for all arms an estimated sign $\hat{s}_k \in \{-1, 1\}$. The objective is to minimize the expected weighted number of missclassified arms, where a mistake on arm $k$ has a known cost $a_k > 0$,

$$L_T = \sum_{k=1}^{K} a_k \mathbb{I}\{\hat{s}_k \neq s_k\} = \sum_{k=1}^{K} a_k E_k . \tag{3}$$

Note that the linear form of the loss is quite general: since $E_k \in \{0, 1\}$, any separable loss $\sum_k f_k(E_k)$ is the sum of a constant and $\sum_k a_k E_k$ for some costs $a_k$.

We conclude this description of the problem with notations used in the design of algorithms. Let $N_{k,t}$ and $\hat{\mu}_{k,t} = \frac{1}{N_{k,t}} \sum_{s=1}^{t} \mathbb{I}\{i_t = k\} X_t$ be the number of times the learner has pulled arm $k$ up to round $t$ (included) and the subsequent empirical mean of arm $k$ repectively. Define further $\hat{\Delta}_{k,t} = |\hat{\mu}_{k,t} - \theta| / \sqrt{2\sigma^2}$ and $\Delta_k = |\mu_k - \theta| / \sqrt{2\sigma^2}$, respectively the empirical and the true (variance-normalized) gap of arm $k$ to the threshold after $t$ rounds.

### 2.1 The benchmarks: a lower bound and a non-adaptive oracle

Following the proof of [35] in a slightly more generic fashion (using exponential families with one parameter instead of Bernoulli distribution), we obtain a lower bound on the performance of any algorithm (see appendix A) from which we get Theorem 1.

**Theorem 1.** *(Similar to Theorem 20 in [35]) Let* $(\Delta_1, \ldots, \Delta_K)$ *be a sequence of gaps. Then for any algorithm and time horizon* $T \geq K$, *there exists an instance in which all arms* $k \in [K]$ *have Gaussian distributions with variance* $\sigma^2$ *and mean in* $\{\Delta_k, -\Delta_k\}$ *such that*

$$\mathbb{E}[L_T] \geq \frac{1}{4} \min_{\sum_k N_k = T} \sum_{k=1}^{K} a_k e^{-4 N_k \Delta_k^2} .$$

We now deriving an optimal but unrealistic oracle, which requires prior knowledge of the gaps as input. Consider the algorithm that pulls each arm $N_{k,T}$ times, a number fixed in advance, then returns the sign of the empirical mean $\hat{\mu}_{k,T}$. Using Hoeffding's inequality, the expected loss verifies:

$$\mathbb{E}[L_T] = \sum_{k=1}^{K} a_k \mathbb{P}\left( (\hat{\mu}_{k,T} - \theta)(\mu_k - \theta) < 0 \right) \leq \sum_{k=1}^{K} a_k e^{-N_{k,T} \Delta_k^2} \tag{4}$$

We define the *non-adaptive oracle* as the allocation $N_T$ which minimizes that upper bound. Its error probability has the same form as the lower bound of Theorem 1, but has a different constant in the exponential (1 instead of 4). We can solve that minimization problem and make the error bound more explicit. To that end, suppose that the arms are ordered such that $a_1 \Delta_1^2 \leq \ldots \leq a_K \Delta_K^2$. There is a set $S = \{k_0, k_0 + 1, \ldots, K\}$ and a constant $C_S$ such that the oracle non-adaptive algorithm has $N_{k,T} = 0$ for $k \notin S$ and $N_{k,T} = \left( C_S + \log(a_k \Delta_k^2) \right) / \Delta_k^2$ for $k \in S$ (see appendix B for details). The expected loss of that non-adaptive oracle is

$$\mathbb{E}[L_T] \leq \sum_{k \notin S} a_k + \sum_{k \in S} a_k \exp\left( -\frac{T + \sum_{j \in S} \frac{1}{\Delta_j^2} \log\left( \frac{a_k \Delta_k^2}{a_j \Delta_j^2} \right)}{\sum_{j \in S} \frac{1}{\Delta_j^2}} \right) . \tag{5}$$

The oracle is not pulling arms $1, \ldots, k_0 - 1$. These are the arms which are too close to the threshold (in a distance weighted by $a_k$) and thus too hard to classify to be worth trying. Giving up on those arms is not something that a non-oracle algorithm can do. Figure 1 illustrates on an example ($\mu_k = (-1)^k (k/K)^2$, $k = 1, \ldots, 50$, and $T = 500$) the shape of the optimal allocation (arms near the threshold should not be drawn) as well as the empirical sampling distributions of several algorithms that pull all arms. In Appendix G, we illustrate how this optimal allocation evolves with the horizon $T$.

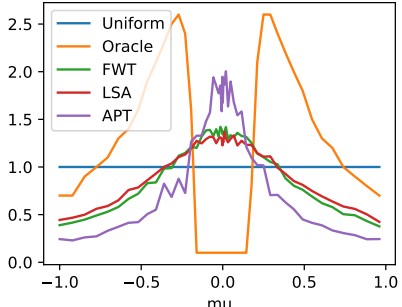

Figure 1: Optimal and empirical sampling distributions with respect to $\mu$.

## 2.2 A good algorithm must pull all arms

We provide a new lower bound for the thresholding bandit with unit-cost problem, to support the claim that it is not possible to avoid pulling the arms which are close to the threshold. Consider the following 4 Gaussian bandit models (with variances 1) with means

$$\mu_{+\varepsilon} = (\varepsilon, \ldots, \varepsilon, \mu_{K_0+1}, \ldots, \mu_K), \qquad \mu'_+ = (\mu_{K_0+1}, \ldots, \mu_{K_0+1}, \mu_{K_0+1}, \ldots, \mu_K),$$
$$\mu_{-\varepsilon} = (-\varepsilon, \ldots, -\varepsilon, \mu_{K_0+1}, \ldots, \mu_K), \qquad \mu'_- = (-\mu_{K_0+1}, \ldots, -\mu_{K_0+1}, \mu_{K_0+1}, \ldots, \mu_K).$$

where $0 < \varepsilon < \mu_{K_0+1} \leq \ldots \leq \mu_K$, the value $\mu_{K_0+1}$ is large enough for the oracle to pull all arms on $\mu'_+$ and $\varepsilon \leq \sqrt{\log(2)/(2T)}$.

**Lemma 1.** If $\mathbb{E}_{\tilde{\mu}}[L_T] \leq c_1 \min_{\sum_k N_k = T} \sum_k e^{-c_0 N_k \Delta_k^2}$ for constants $c_0, c_1$ on $\tilde{\mu} \in \{\mu'_+, \mu'_-\}$, then

$$\max_{\mu \in \{\mu_{+\varepsilon}, \mu_{-\varepsilon}\}} \mathbb{E}_\mu \left[ \sum_{k=1}^{K_0} N_{k,T} \right] \geq \frac{1}{2(\mu_{K_0+1} - \varepsilon)^2} \left( c_0 \frac{T + H^{\log}}{H} + \log \frac{K_0}{32 c_1 H} \right).$$

where $H = \frac{K_0}{\Delta_{K_0+1}^2} + \sum_{k=K_0+1}^K \frac{1}{\Delta_k^2}$ and $H^{\log} = \frac{K_0}{\Delta_{K_0+1}^2} \log \frac{1}{\Delta_{K_0+1}^2} + \sum_{k=K_0+1}^K \frac{1}{\Delta_k^2} \log \frac{1}{\Delta_k^2}$.

The proof is postponed to Appendix A. In a few words, if an algorithm has an expected loss close to the loss of the non-adaptive oracle, then it must pull linearly the arms which are close to the threshold.

# 3 Algorithm and upper-bound

We introduce and analyse a new class of algorithms for the thresholding bandit problem that we call *index-based* algorithms. That class unifies several existing algorithms, including APT [28] and LSA [35]. An index-based algorithm pulls the minimum of $K$ quantities, one for each arm, that each depends only on the rewards and pull counts of the respective arm (it does not change when pulling other arms). In particular, we consider algorithms for which the sampled arm is $i_{t+1} \in \arg\min_{k \in [K]} F(N_{k,t}, N_{k,t} \hat{\Delta}_{k,t}^2; a_k)$ for a function $F : \mathbb{N} \times \mathbb{R}_+ \times \mathbb{R}_+^* \to \mathbb{R}$ that depends on the pull counts, the information about the sign and the weight of the arm.

After $T$ rounds, the algorithm recommends the sign of the arms at the round $t_{\max} \in [T]$ at which $\min_{k \in [K]} F(N_{k,t}, N_{k,t} \hat{\Delta}_{k,t}^2; a_k)$ was maximal. This rule is used as opposed to returning the sign of all arms at time $T$ to facilitate the analysis, which is based on the observation that there is a small probability of error when all arms have high index. The time $t_{\max}$ should be close to $T$: in particular, only one arm is sampled (possibly several times) between $t_{\max}$ and $T$ (see Appendix C). In Sec. 3.2, we provide a generic analysis for index-based algorithms satisfying the assumption below.

**Assumption 1.** The index function $F(n, x; a) : \mathbb{N} \times \mathbb{R}_+ \times \mathbb{R}_+^* \to \mathbb{R}$ is non-decreasing in $n$ and $x$ and $\lim_{n \to +\infty} F(n, ny; a) = +\infty$ for all $y > 0, a > 0$.

Intuitively, algorithms that verify Assumption 1 prefer pulling arms that were pulled the least (smallest $n$) and whose quantity of information about the sign ($n\hat{\Delta}_{k,n}^2$) is small. This class includes several algorithms from the thresholding bandits literature: APT [28] for $F(n, x; a_k) = x$ and LSA [35] for $F(n, x; a_k) = x + \log(n)$ (these algorithms are only defined for $a_k = 1$). We now propose a generic method for designing an index-based algorithm.

**Algorithm 1** Index-based algorithm for thresholding bandit

**Inputs**: an index function $F : \mathbb{N} \times \mathbb{R}_+ \times \mathbb{R}_+^* \to \mathbb{R}$; $a_1, \ldots, a_K \in \mathbb{R}_+^*$; $\sigma > 0$; and $\theta \in \mathbb{R}$

For $t = 1, \ldots, T$ do

- for all $k \in [K]$ define

$$N_{k,t-1} = \sum_{s=1}^{t-1} \mathbb{I}\{k = i_s\}, \quad \hat{\mu}_{k,t-1} = \frac{1}{N_{k,t-1}} \sum_{s=1}^{t-1} \mathbb{I}\{k = i_s\} X_s, \text{ and } \hat{\Delta}_{k,t-1}^2 = \frac{1}{2\sigma^2}\left(\hat{\mu}_{k,t-1} - \theta\right)^2$$

- pull $i_t \in \arg\min_{k \in [K]} F\left(N_{k,t-1}, N_{k,t-1}\hat{\Delta}_{k,t-1}^2; a_k\right)$.
- observe $X_t \sim \nu_{i_t}$

Define $t_{\max} = \max_{t \in [T]} \min_{k \in [K]} F\left(N_{k,t}, N_{k,t}\hat{\Delta}_{k,t}^2; a_k\right)$

Return for each $k \in [K]$ the sign $\hat{s}_k = \mathrm{sign}(\hat{\mu}_{k,t_{\max}} - \theta)$

---

### 3.1 Frank-Wolfe for Thresholding bandits

Our strategy to minimize the expected loss is inspired by the Frank-Wolfe algorithm [15] and aims at controlling an upper-bound on the loss, such as the right hand side of Inequality (4). Let's write that function as $B(N_T) = \sum_{k=1}^{K} a_k e^{-N_{k,T}\Delta_k^2}$. The high-level idea is to sequentially estimate its gradient and move to the minimizer of its linear approximation. If the gaps were known, we could compute at time $t+1$ the gradient of the bound with respect to $N_t$, $\nabla B(N_t) = (-a_k\Delta_k^2 e^{-N_{k,t}\Delta_k^2})_k$ and use the Frank-Wolfe algorithm. The algorithm would pull $i_{t+1} \in \arg\min_u u^\top \nabla B(N_t)$ for $u$ in the simplex, which is simply $\arg\min_{k \in [K]} (-a_k\Delta_k^2 e^{-N_{k,t}\Delta_k^2})$. The gaps are however unknown. We therefore compute an estimate of the gaps $\hat{\Delta}_{k,t}$, with which we form the estimated gradient

$$\hat{\nabla} B(N_t)_k = -a_k\hat{\Delta}_k^2 e^{-N_{k,t}\hat{\Delta}_{k,t}^2} = -\exp\left(-\left(N_{k,t}\hat{\Delta}_{k,t}^2 - \log(N_{k,t}\hat{\Delta}_{k,t}^2) + \log\left(\frac{N_{k,t}}{a_k}\right)\right)\right).$$

This gives a natural choice for the index function of our algorithm $F(n, x; a_k) = x - \log x + \log(n/a_k)$. However, the latter is decreasing in $x$ for $x \in (0, 1)$, which in addition to violating Assumption 1, may lead to instability in the initial phase when the gaps $\Delta_k$ are poorly estimated by $\hat{\Delta}_{k,n}$. We therefore propose a slight modification that preserves the asymptotic behavior of $F$ and we call the resulting algorithm FWT (Frank-Wolfe for Thresholding bandits):

$$F(n, x; a_k) = \max\{x, 1\} - \log(\max\{x, 1\}) + \log(n/a_k). \tag{FWT}$$

**Recovering APT** Using different upper-bounds $B$ on the expected loss may lead to different algorithms. In particular, we highlight a link between our Frank-Wolfe inspired method and the APT algorithm of [28], which was designed to minimize the loss

$$L_T = \sum_{k=1}^{K} a_k \mathbb{I}\{\hat{s}_k \neq s_k\} = \sum_{k=1}^{K} a_k E_k.$$

Following our method with the choice $B(N_t) = \max_{k \in [K]} e^{-N_{k,t}\Delta_k^2}$ results in exactly the same sampling rule as the one of the APT algorithm (the recommendation rule differs slightly since we recommend the sign at $t_{\max}$ and not at $T$). Indeed, the derivative of $B$ with respect to $N_{k,t}$ is nonzero (and negative) if and only if $N_{k,t}\Delta_k^2 = \arg\min_j N_{j,t}\Delta_j^2$ (ignoring the case in which there are several argmins, for which the tie breaking can be arbitrary). This leads to the choice $F(n, x; a_k) = x$ in Algorithm 1, which then pulls $i_{t+1} = \arg\min_{k \in [K]} N_{k,t}\hat{\Delta}_k^2$. This is the sampling rule of APT.

### 3.2 Loss upper bound

We provide a loss upper bound that is valid for all index-based algorithms that verify Assumption 1. We then give a compact summary of the analysis outline and the resulting loss bounds.

**Theorem 2.** *Let $K \geq 1$, $a_1, \ldots, a_K > 0$, $T \geq 1$, and $\sigma > 0$. Let $F : \mathbb{N} \times \mathbb{R} \times \mathbb{R}_+^* \to \mathbb{R}$ that satisfies Assumption 1. Let $C_1, \ldots, C_K > \max_k F(0, 0; a_k)$. For all $j, k \in [K]$, define*

- $t_j(C_k)$ *a solution of the equation* $F(t, t\Delta_j^2; a_j) = C_k$,
- $S_k \subseteq [K]$ *and* $t_{j,0}(C_k) \in \mathbb{R}_+$, *a set and values such that for* $i \notin S_k$,
  $\mathbb{P}\left(\exists n \leq t_{i,0}(C_k), F(n, n\hat{\Delta}_{n,i}^2; a_i) \geq C_k\right) = 1$.

*Then the expected loss of Algorithm 1 is upper-bounded as*

$$\mathbb{E}[L_T^{\mathbb{A}}] \leq \sum_{k=1}^{K} a_k \left( e \cdot \exp\left( -\frac{\frac{1}{2}\left(T - \sum_{j \notin S_k} t_{j,0}(C_k)\right) - \sum_{j \in S_k} t_j(C_k)}{\sum_{j \in S_k} 1/\Delta_j^2} \right) + T \cdot e^{-t_k(C_k)\Delta_k^2} \right).$$

Refer to Appendix D for the proof. It is composed of two parts:

1. First we establish that for any arm $j \in [K]$, with large probability, there is a time $\tau_j(C_k)$ such that $F(\tau_j(C_k), \tau_j(C_k)\hat{\Delta}_{\tau_j(C_k),j}; a_j) \geq C_k$. We prove that for all $j, k \in [K], \tau_j(C_k)$ has an exponential tail then use the fact that the algorithm pulls the minimal index to control the probability that the minimum never reaches $C_k$.

2. We show that if an arm's index is large, then the probability of mistake on it is small.

The times $t_j(C_k)$ of Theorem 2 are the smallest numbers of samples such that $t_j(C_k) \geq \tau_j(C_k)$ with high enough probability. By determining those times, we derive explicit bounds for algorithms that verify Assumption 1. In particular we derive a bound for the variant of APT which returns the sign at the time $t_{\max}$ when the minimal index was maximal.

**Corollary 1.** Suppose that for all $k \in [K]$, $a_k = 1$. For all $T \in \mathbb{N}^*$,

$$\mathbb{E}[L_T^{\text{APT}}] \leq 2K\sqrt{e \cdot T} \cdot \exp\left( -\frac{1}{4}\frac{T}{\sum_{j=1}^{K} 1/\Delta_j^2} \right).$$

Refer to Appendix D.3 for the proof. Since $\max_k E_k \leq \sum_k E_k$, the bound of Corollary 1 is also a bound on the zero-one loss, which we can compare to the result of [28]. Our result shows a $1/4$ factor in the exponential instead of the worse $1/32$ constant of the original paper.

**LSA and FWT**    Theorem 2 applies to LSA and FWT with the following times:
- LSA: $t_j(C_k) = W(e^{C_k}\Delta_j^2)/\Delta_j^2$ and $t_{j,0}(C_k) = e^{C_k}$,
- FWT: $t_j(C_k) = \log(e^{C_k}a_j\Delta_j^2)/\Delta_j^2$ and $t_{j,0}(C_k) = a_j e^{C_k - 1}$,

where $W$ is the Lambert W function, which verifies $|W(x) - (\log x - \log\log x)| \leq \log(1 + 1/e)$ for $x \geq e$. Therefore, for the two algorithms, the times $t_j(C_k)$ are close (equal up to the $\log\log$ terms in $W$), thus their bounds are close as well. Note that LSA is only defined for $a_j = 1$ for all $j$. In contrast to LSA, our bound for FWT has the notable property that, in the regime where $T \geq 2\sum_{j=1}^{K} \frac{1}{\Delta_j^2}(2 + \log\frac{a_j\Delta_j^2 \max_i a_i\Delta_i^2}{(\min_k a_k\Delta_k^2)^2} - \log\frac{T}{e})$, we recover the same exponent as in the non-adaptive oracle loss bound (5) (up to a factor $1/4$). Indeed we show that for such $T$

$$\mathbb{E}[L_T^{\text{FWT}}] \leq 2\sqrt{eT}\sum_{k=1}^{K} a_k \exp\left( -\frac{1}{4}\frac{T + 2\sum_{j=1}^{K}\frac{1}{\Delta_j^2}\log\frac{a_k\Delta_k^2}{a_j\Delta_j^2}}{\sum_{j=1}^{K} 1/\Delta_j^2} \right). \tag{6}$$

In the same regime of large $T$, the bound that we obtain for LSA is of the same order, but less explicit due to the function $W$. The latter is still impressive since the original theorem of [35] for LSA exhibits an exponent significantly looser, of order $\exp\left( -\frac{1}{16020}\frac{T}{\sum_{j=1}^{K} 1/\Delta_j^2} \right)$, *i.e.* 4005-times worse than our bound. We finally derive a bound for our newly introduced algorithm.

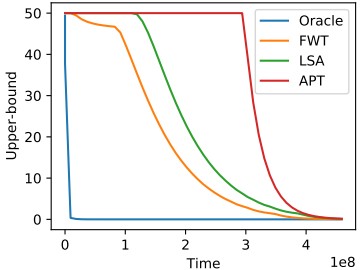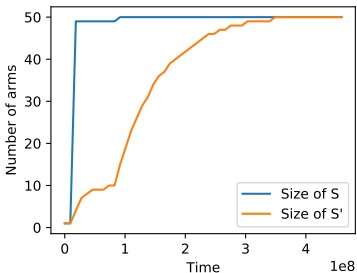

Figure 2: [left] Comparison of the upper-bounds of Corollaries 1, 2 and that of the optimal non-adaptive oracle of Eq. (5) (blue) when the gaps are of the form $\Delta_i = (i/K)^2$. [right] Evolution over time of the size of the optimal sets $S$ (blue) and $S'$ (orange) that minimize the bound of Corollary 2.

**Corollary 2.** Let $S, S'$ be two sets with $S' \subseteq S \subseteq [K]$ and let $C \in \mathbb{R}$ be such that $C \geq 1 + \max_{k \in S} \log \frac{1}{a_k \Delta_k^2}$. Then, for all $T \geq 1$

$$\mathbb{E}[L_T^{\text{FWT}}] \leq \sum_{k \notin S'} a_k + e \sum_{k \in S'} a_k \exp\left( -\frac{\frac{1}{2}(T - \sum_{j \notin S} a_j e^{C-1}) + \sum_{j \in S} \frac{1}{\Delta_j^2} \log \frac{1}{a_j \Delta_j^2}}{\sum_{j \in S} 1/\Delta_j^2} + C \right)$$
$$+ T \sum_{k \in S'} a_k \exp\left( -C + \log(1/(a_k \Delta_k^2)) \right) .$$

Figure 2 compares the upper-bounds of Corollary 1 (APT), Theorem 2 (see also Equation (9) in the Appendix) (LSA), and Corollary 2 (FWT) for the particular case $\Delta_i = (i/K)^2$ and $a_i = 1$, for $i = 1, \ldots, K = 50$. See also Figure 4 in the supplementary material for $\Delta_i = i/K$. We can see that while the bounds of LSA, APT, and FWT are asymptotically similar, that of FWT starts to be significant for much smaller $T$. On the right, we can see the importance of the set $S'$ in Corollary 2: the bounds first ignores all the arms, and suffers a loss of 1 and then adds them one by one as soon as they can be classified. The bound derived in [35] for LSA is not represented on the figures, since it is still bigger than $K$ for the considered range of $T$.

### 3.3 The sum-of-gaps objective

We show that our method applies for the sum-of-gaps objective $\sum_{k=1}^{K} \Delta_k E_k$. This is not a particular case of the setting discussed previously since $a_k$ was known to the algorithm, while $\Delta_k$ is unknown. It serves as a proof of concept for the extensibility of our method. The index given by FWT in this setting is $F(n, x) = x' - \frac{3}{2} \log (x') + \frac{3}{2} \log (n)$, where $x' = \max\left(x, \frac{3}{2}\right)$. We can then bound the sum-of-gaps loss using our generic analysis by proceeding similarly to Theorem (2).

**Corollary 3. (FWT for the sum-of-gaps objective)** In the regime where $T \geq 2 \sum_{j=1}^{k} \frac{1}{\Delta_j^2} \left( 3 + 3 \log \frac{\Delta_j \max_i \Delta_i}{(\min_i \Delta_i)^2} - \log \frac{T}{e} \right)$, we show that

$$\mathbb{E}[\sum_{k=1}^{K} \Delta_k E_k] \leq 2\sqrt{eT} \sum_{k} \Delta_k \exp\left( -\frac{1}{2} \frac{\frac{T}{2} + \sum_j \frac{3}{2} \frac{1}{\Delta_j^2} \log \frac{\Delta_k^2}{\Delta_j^2}}{\sum_j 1/\Delta_j^2} \right) .$$

See Appendix E for the proof and for a different bound that is valid for all times $T$. This can be useful for applications in which errors are more tolerated for arms that are close to the threshold.

## 4 Beating the oracle? The benefits of adaptivity.

We argue that in some situations adaptive algorithms can greatly outperform the non-adaptive oracle of Section 2.1, i.e., the cost of non-adaptivity can be much higher than the cost of learning. The algorithms in the family we considered are all adaptive in the sense that they adapt their drawing strategy as more information is observed, at the cost of learning the parameter $\mu_k$. We illustrate the benefits of adaptivity in the following toy example.

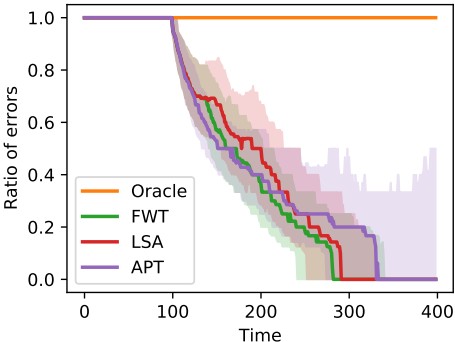 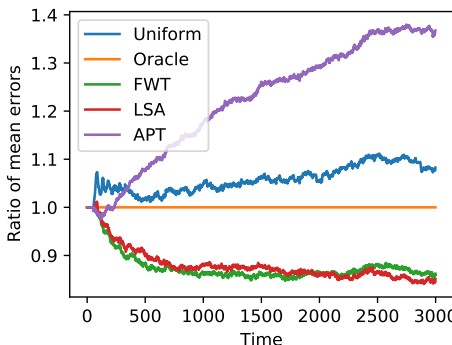

Figure 3: [left] Median (and $0.25$, $0.75$ empirical quantiles obtained on $500$ runs) of the ratio between the error suffered by each algorithm and that of the optimal non-adaptive oracle ($\mu_k = (-1)^k, k = 1, \ldots, 100$). [right] Ratio of the averaged errors (over $500$ runs) of each algorithm with that of the oracle ($\mu_k = (-1)^k(k/K)^2, k = 1, \ldots, 50$).

**The "optimal" non-adaptive algorithm may be worse than adaptive algorithms.** Consider the following parametric problem. An arm distribution is parametrized by $x \in \mathbb{R}$ and is supported on $\{0, x\}$; a sample of that distribution is equal to $0$ or $x$, each with probability $1/2$. We assume that all arms have non-zero parameter and we will compute the optimal non-adaptive allocation.

We make the convention that if an algorithm sees only zeros for one arm, it returns any sign with probability $1/2$. The error probability of a non-adaptive allocation $N_T^k$ for arm $k$ is half of the probability of seeing only zeros (since if anything else is observed, the arm can be classified with perfect accuracy). Hence the total error is

$$\mathbb{E}\big[L_T\big] = \frac{1}{2} \sum_{k=1}^{K} \frac{1}{2^{N_{k,T}}} \geq \frac{K}{2^{(T/K)+1}} \;,$$

which is minimized with the uniform allocation: $N_{k,T} = \frac{T}{K}$ for all $k \in [K]$.

Consider now an adaptive procedure that sample each arm in turn, but stops sampling an arm as soon at it sees a non-zero value. We crudely prove an upper bound for its number of errors, by remarking that it is zero if the algorithm classifies all arms correctly and smaller than $K$ otherwise. The number of samples required to perfectly classify an arm follows a geometric distribution with parameter $1/2$. As a consequence, the number of required samples to classify all arms correctly follows a negative binomial $\text{NB}(K, 1/2)$. Let $Z$ be such a negative binomial random variable. The expected number of errors of the adaptive procedure is up to $K\mathbb{P}(Z > T)$. It then verifies

$$\mathbb{E}\big[L_T\big] \leq K\mathbb{P}(Z > T) \leq Ke^{-(\log(2)/2)T}\mathbb{E}e^{(\log(2)/2)Z} = \frac{K}{2^{T/2}} \left(1 + \frac{1}{\sqrt{2}}\right)^K \;,$$

where the value $\log(2)/2$ is chosen for simplicity (in $[0, \log 2)$). In the regime where $T$ is large, this is of order $1/2^{T/2}$, which for $K > 2$ is much smaller than $1/2^{T/K}$ for the uniform allocation.

This toy example differs drastically from more realistic situations, as one non-zero sample for an arm is sufficient to know the sign of the expectation perfectly. We therefore consider empirically more reasonable frameworks, closer to those analyzed in the paper: the distributions of $K$ arms are either $\mathcal{N}(1, 1)$ or $\mathcal{N}(-1, 1)$. Since all gaps $\Delta_i$ are equal, the optimal non-adaptive oracle is also the uniform sampling. The results are illustrated on the left part of Figure 3 and highlight the fact that all the adaptive algorithms considered (APT, LSA or FWT) drastically outperform the oracle. The right part of the figure shows the same phenomenon on another example in which the gaps are not constant. In particular, we can see that FWT and LSA have similar performance while APT (not designed for this purpose) generally suffers from a larger error. This result was corroborated by most of our experiments. We refer to Appendix G for more details.

## Discussion

An interesting research direction is to consider objective functions more general than (1). In particular, we believe that our approach can be generalized to losses of the form $L_T = \sum_{k=1}^{K} f(\Delta_k, E_k)$ under certain regularity assumptions on $f$. Moreover, we focused on separable losses (hence linear wlog) and the index based algorithms we analyze reflect that separability. An obvious and intriguing direction for further work is to replace that assumption. One might for example want to design an algorithm that minimizes the probability of making more than a given number of mistakes.

The fact that adaptive algorithms can beat non-adaptive oracles has already been observed empirically for fixed confidence identification [33, 10], although only in cases where the non-adaptive oracle was worse only for small times and was still asymptotically optimal. The phenomenon we observe for fixed budget thresholding is much more significant and remains to be explained by theoretical arguments. Currently, the best theoretical bound for adaptive algorithms is still a factor $1/4$ away in the exponent from the non-adaptive oracle bound.

## Acknowledgments and Disclosure of Funding

V. Perchet acknowledges support from the French National Research Agency (ANR) under grant number #ANR-19-CE23-0026 as well as the support grant, as well as from the grant "Investissements d'Avenir" (LabEx Ecodec/ANR-11-LABX-0047)". R. Ouhamma also awknowledges support from Ecole polytechnique under the AMX funding. P. Gaillard and R. Degenne were supported by the French government under management of Agence Nationale de la Recherche as part of the "Investissements d'avenir" program, reference ANR-19-P3IA-0001 (PRAIRIE 3IA Institute).

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
