# OpenReview forum: "Online Sign Identification: Minimization of the Number of Errors in Thresholding Bandits"
_NeurIPS.cc/2021/Conference — Accept (Spotlight)_

### Official Review · Reviewer_JWse · 2021-07-05

**Rating:** 7
**Confidence:** 4

**Summary:**

This paper studies the thresholding bandit problem with the goal which can be seen as a weighted version of aggregate regret. The main contributions of this paper are two-fold:

1. It introduces an algorithm that is inspired by the Frank-Wolfe algorithm regarding this generic goal. A corresponding upper bound is derived regarding this algorithm.
2. This paper discovers new insights on the thresholding bandit problem i.e., a good algorithm must pull all arms and the “optimal” non-adaptive algorithm may be worse than adaptive algorithms.

After rebuttal
--------------------
The rebuttal looks good to me and I will keep my positive score.

**Main Review:**

Generally, I think this is a good paper with a nice literature review, presentation, and technique depth. The developed new algorithm, which aims to solve a generic goal, is found to have a better regret bound than the previous LSA algorithm when considering the goal to minimize aggregate regret. Also, it is interesting to see the new rule to decide the signs of the arms.

I do have some minor comments detailed in the following:

Line 36: “As as consequence” -> As a consequence

Line 49: “the performances of an algorithm”: performances -> performance

Lemma 1: what is \tilde{\Delta}_k^2?

Line 179: “that each depend only on the rewards”: depend -> depends

Assumption 1: it is better to specify the parameters of F i.e., F(n, x; a) for clearer presentation

Line 278: in the example introduced, the sign is not defined. I guess the sign should be the actual parameter x?

Figure 3: In the caption of the left subfigure, it says Median, 0.25, and 0.75 empirical quantiles are presented. However, for each algorithm, only one line is presented. I am wondering which one you are referring to?  Also for the right subfigure, the ratio of the averaged errors should be at most 1, and why it appears to be greater than 1 in the figure?

Regarding the experiments, I think it may be better to add one more figure presenting the performance of algorithms LSA, APT, and FWT regarding random input instances. The current one \mu_k = (-1)^k (k/K)^2 looks artificial to me.

**Time Spent Reviewing:**

2

---

> ### Author Response · Authors · 2021-08-10
> **Authors response**
>
> We thank the reviewer for the nice remarks about the novelty and generality of our work.
> We now address specific comments:
>
> "Lemma 1: what is $\tilde{\Delta}_k^2$?"
>
> It's a typo and should be replaced by $\Delta_k^2$, the squared gap.
>
> "In the caption of the left subfigure, it says Median, 0.25, and 0.75 empirical quantiles are presented. However, for each algorithm, only one line is presented."
>
> The line is the median, and the quantiles are represented by the shaded area of the same color.
>
> "Also for the right subfigure, the ratio of the averaged errors should be at most 1, and why it appears to be greater than 1 in the figure?"
>
> The ratio can be higher than 1 if the algorithm makes more mistakes than the non-adaptive oracle. It is the case for APT (which is not designed for that task) and for the uniform allocation, which is not the optimal non-adaptive allocation for that example. In contrast, the adaptive algorithms LSA and FWT which are designed for the correct task have average error lower than the non-adaptive oracle.
>
> Note also that the two sub-figures of figure 3 correspond to experiment with different arm means. The error proportions of all algorithms are lower than that of the oracle on the left experiment (in which APT happens to be a sensible algorithm), but not on the right.

---

### Official Review · Reviewer_ydCQ · 2021-07-16

**Rating:** 7
**Confidence:** 4

**Summary:**

This paper works on the problem of fixed-budget sign identification problem in stochastic bandits. Given a set of arms where each sample on an arm returns a noisy result according to this arm's unknown latent distribution, a threshold, and the limited time horizon $T$, the learner needs to adaptively sample the arms and wants to minimize the error probability of determining whether the mean rewards of the arms are above or below the threshold.

The authors did a thorough and careful analysis on the performance of a large class of algorithms as well as the lower bound of this problem. By the analysis, the authors managed to get a gap 4 in the exponential part of the upper bound and the lower bound of the error probability, outperforming the existing results.

The results in this paper help the researchers to understand the problem of online fixed-budget sign identification problem, and the new bounds are promising both in theory and in practice.

**Limitations And Societal Impact:**

The limitations have been discussed in the box Main Review.

**Main Review:**

The problem studied in this paper is well-known in the community. The algorithms are not very novel according to my knowledge. However, the authors managed to get better bounds by a better analysis. Another main contribution of this paper is to make an abstraction of the existing algorithms and make a unified and thorough analysis.

The results are very close to the optimal, i.e., there are only a gap of 4 in the exponential term.

The paper is well written and well structured.

The main concern of this paper is the significance. In fact, this paper does not propose novel algorithms or mathematical techniques, but more like a deeper study and summary of existing works. But due to its quality and clarity, I tend to give a score between 6 and 7. To be more conservative in case I missed anything, I will vote a 6 for now and wait for the author responses.

--------------------------------------
After the rebuttal, I am raising the score from 6 to 7.

**Time Spent Reviewing:**

2

---

> ### Author Response · Authors · 2021-08-10
> **Authors response**
>
> We thank the reviewer for the positive comments on our analysis of the algorithms.
> The main concern expressed in the review is the significance of our results, which we address in two parts. First, we object to the affirmation that "this paper does not propose novel algorithms or mathematical techniques". Second, our paper brings more hindsight about the problem, beyond loss bounds.
>
> Out of the three algorithms that we analyse, one is new (FWT). That algorithm enjoys the tightest upper bound of the three. Although it is based on the idea of the well known Frank-Wolfe method, the design of FWT requires additional ingredients to obtain an index based algorithm with the desired properties.
> Furthermore, as pointed out in the review, the way we analyse all three methods is based on a new abstraction using generic index based algorithms. This, together with the more minor innovation of returning the sign of the arm at the time at which the index is maximal (and not at the final time), allows us to derive tight bounds for all algorithms with one unified proof.
> Section 3.3 shows that the FWT algorithm and the analysis are general and extend to more tasks, like the sum-of-gaps objective.
>
> Beyond proving error bounds for fixed budget problems, we also highlight in section 4 the benefits of adaptivity. Adaptive algorithms beat the oracle empirically and on a theoretical toy example that we introduce. Since the state of the art for upper bounds don't even show that adaptive algorithms match the oracle, this is an intriguing observation about fixed budget identification.

---

> > ### Comment · Reviewer_ydCQ · 2021-08-20
> > **Raising the scores after reading the response and other authors' reviews**
> >
> > Thank you for the response. Also, I have read the reviews of other reviewers. I am convinced that the results of this paper are of reasonable significance so I will raise my score to 7.

---

### Official Review · Reviewer_9Lq4 · 2021-07-21

**Rating:** 8
**Confidence:** 4

**Summary:**

The authors come up with a general way to analyze algorithms that solve the thresholded bandit problem. Given any function that trades-off less sampled arms and those close to the threshold. They show that some existing algorithms fit their framework (LSA and APT). They give new lower bounds and show that their adaptive algorithm outperforms a non-adaptive oracle that is aware of all the gaps.

Their main contributions are:
-  a new proof technique for thresholded bandits
- new upper and lower bounds
- proof that the adaptive method can beat a non-adaptive oracle in some cases

**Limitations And Societal Impact:**

The authors have addressed the limitations but I did not find their note on social impact. Having said that, this work is mostly theoretical posing a difficulty to analyze its societal impact.

**Main Review:**

Originality: their proof based on the Frank-Wolfe method for any index function that satisfies their criteria is novel. It encompasses current methods but they also provide a better algorithm (for the constant in the exponent).

Clarity: The paper is well written. This reviewer found a few typos (see below) but it is clear otherwise.
 Significance: the results are useful because they give a general proof method that may be applied to other algorithms. It provides a structure to construct good index functions for thee bandits based on their bounds. I think that this paper will help other build on the results provided.

Critiques: the authors give an example of when adaptivity helps but they also show cases where it hurts (non-oracle algorithms cannot ignore certain arms like the oracle). This is an interesting point and a discussion about what types of problems fall into each category would have enhanced the paper.

**Time Spent Reviewing:**

12

---

> ### Author Response · Authors · 2021-08-10
> **Authors response**
>
> We thank the reviewer for the time spent on the review and the positive feedback. We agree that characterizing the tasks on which adaptivity helps is a very interesting research direction. In this work, we managed to exhibit experimentally examples of tasks in which it helps, as well as a theoretical toy example. We are not yet in a position to give a more general characterization of when an adaptive algorithm can beat the non-adaptive oracle. A next step could be to obtain upper bounds for the adaptive algorithms that actually match (or even beat) the performance of the oracle, by extending the proof method of the toy example of section 4: an adaptive algorithm can "gain samples" by using the observations to make early decisions when the random deviations are favorable.

---

### Decision · Program_Chairs · 2021-09-27

**Decision:**

Accept (Spotlight)

**Comment:**

The reviewers came to consensus that this paper makes a good progress on the thresholding bandit problem from the perspectives of upper and lower bounds as well as an interesting discussion on the benefit of adaptivity. I agree with these opinions and please polish the manuscript so that the minor concerns raised by the reviewers become clear in the final version.